# Apprehensions and Optimism among Dental Professionals during the COVID-19 Pandemic—A Nationwide Cross-Sectional Evaluation

**DOI:** 10.3390/vaccines10122081

**Published:** 2022-12-06

**Authors:** Lakshmi Nidhi Rao, Aditya Shetty, Priyanka Latha Senthilkumar, Prasanna Kumar J Rao, Heeresh Shetty, Shreya K Shetty, Vidya G Doddawad, Srikant Natarajan, Ajinkya M Pawar, Manjeshwar Shrinath Baliga, Alexander Maniangat Luke, Mohmed Isaqali Karobari

**Affiliations:** 1Department of Conservative Dentistry and Endodontics, AB Shetty Memorial Institute of Dental Sciences, NITTE Deemed to be University, Mangalore 575018, Karnataka, India; 2Department of Conservative Dentistry and Endodontics, Vivekanandha Dental College for Women, Tiruchengode 637205, Tamilnadu, India; 3Department of Oral Medicine and Radiology, A.J. Institute of Dental Sciences, Mangalore 575004, Karnataka, India; 4Department of Conservative Dentistry & Endodontics, Nair Hospital Dental College, Mumbai 400008, Maharashtra, India; 5Department of Orthodontics, Government Dental College and Research Institute, Bangalore 560002, Karnataka, India; 6Department of Oral Pathology and Microbiology, JSS Dental College and Hospital, Mysore 570015, Karnataka, India; 7Department of Oral Pathology and Microbiology, Manipal College of Dental Sciences, Manipal Academy of Higher Education, Mangalore 575001, Karnataka, India; 8Research Unit, Mangalore Institute of Oncology, Pumpwell, Mangalore 575002, Karnataka, India; 9Department of Clinical Science, College of Dentistry, Ajman University, Al-Jurf, Ajman 346, United Arab Emirates; 10Center of Medical and Bio-Allied Health Sciences Research, Ajman University, Al-Jurf, Ajman 346, United Arab Emirates; 11Department of Restorative Dentistry & Endodontics, Faculty of Dentistry, University of Puthisastra, Phnom Penh 12211, Cambodia; 12Conservative Dentistry & Endodontics, Saveetha Dental College & Hospitals, Saveetha Institute of Medical and Technical Sciences University, Chennai 600077, Tamil Nadu, India

**Keywords:** COVID-19, pandemic, dental professionals, impact on practice, disease transmission

## Abstract

Globally, the pandemic of the coronavirus disease, which started in Wuhan, China, has become a major issue for public health. The COVID-19 epidemic notably causes health professionals to experience significantly more emotional stress than the general populace. The present study proposes to investigate the fear aspect in dentists in the initiation of clinical practice during these times. An online cross-sectional study was conducted among dental practitioners based on a pre-validated questionnaire. The data were expressed as frequency and percentage analyzed using the chi-square test using SPSS version 25. The data was obtained from 271 participants, where clinical practice after the first wave was mostly by freelancers (*p*-value = 0.01); most of whom were married (*p*-value = 0.065); 19.1% attached to institutes did not have changes in earnings; 28.1% of private practitioners had less than 10% reduction in cases. A total of 62% of private practitioners are concerned about the vaccine’s preventative effects (*p*-value = 0.026), and 57% of private practitioners worry about being sued for the delay in treatment (*p*-value = 0.036). Only 33.1% of employees in institutions worry that becoming sick could endanger their family. As dentists continue to work their way through this pandemic, these pressures only occasionally have an impact on them. According to the researchers, this is the first study that has captured the anxiety and apprehensions that dental practitioners experienced during the height of the COVID-19 outbreak in India in April 2021. The study’s findings demonstrate that the study population was generally upbeat and confident that they could quickly overcome their fear.

## 1. Introduction

Ahead of the announcement of the pandemic of coronavirus disease (COVID-19) which started in Wuhan, China in December 2019, every country in the world has repeatedly implemented harsh restrictive measures focused mostly on social isolation, even with nearly complete population confinement, with significant ongoing personal, social, and economic repercussions [1]. The COVID-19 disease has become a major concern for public health [2]. The effects could be detrimental to mental health and have an impact on psychological well-being. It is crucial to reckon with any potential global mental health problems they may cause. Financial problems, fear of infection and mortality, social isolation and lockdown, and behavioral, physical, and psychological symptoms brought on by mental health impairment were the significant issues during the COVID-19 pandemic [2,3,4,5]. Specifically, women are more frightened of and worried about COVID-19 than men. Women are more concerned about COVID-19’s potential negative health effects than men, despite the fact that men are more likely to encounter these effects than women [6].

With the subsequent nation-wide lockdown in India, only critically ill individuals were given priority for medical care during this period, which largely restricted the spread of the infection. Due to the prevalence of widespread infection and long-standing non-critical but significant medical treatments, especially dental ones, the second wave of the COVID-19 outbreak in early 2021 caused an unprecedented disaster in India. Dentists were classified as having a very high exposure risk during this period with a high potential to be infected with the virus [7].

Based on genetic and epidemiological studies, COVID-19 appears to have begun with a single animal-to-human transmission accompanied by an ongoing human transmission [8]. While the principal cause of transmission was patients with symptoms of COVID-19, research findings indicate that asymptomatic patients and patients at the time of the incubation did carry SARS-CoV-2 [9].

During a pandemic, some dentists may have no option but to treat emergency patients, with conditions such as such as pain, bleeding, and sepsis [10]. The standard protective measures in daily clinical work are not effective enough to prevent the spread of COVID-19, especially in patients in incubation, unaware of their being infected, or choosing to cover their infection, given the unique characteristics of a dental procedure that could produce many droplets or aerosols [11].

With world news stressing the strict steps to monitor personal hygiene needed to tackle the transmission of COVID-19, worldwide people are more aware of the importance of attending medical centers, including dental clinics, regardless of their treatments (restorative, endodontics, or dental extractions). To prevent COVID-19 infection from being contracted in the community and/or other humans, the number of patients visiting dental clinics has significantly decreased worldwide. It was observed that during the COVID-19 pandemic, the health workers encountered much more emotional stress than the general public [12]. HCWs reported moderate to severe depression in percentages ranging from 12.1% to 50.4%. In a study that examined stress, anxiety, depression, and sleepiness, 63% of healthcare professionals stated that they had mental disturbance [13]. Additionally, 39.1% of participants in the study by Dai et al. had GHQ-12 scores below 3, which was substantially above average [14]. In another study, an astonishingly increased incidence of HCWs expressing anxiety, sadness, and stress at rates of 67.55%, 55.89%, and 62.99%, respectively, was reported [15]. Furthermore, healthcare professionals (nurses, front-line, and younger medical staff) reported increased severe levels of all psychiatric conditions excluding vicarious traumatization. The severity of patients, working hours per week, food, and sleep quality also affected HCWs’ stress levels, as did the fact that they were the only children in their homes [14,16,17].

The major problems of health personnel during pandemics have been increased workload, working with repeatedly changing protocol, using PPE, social distancing, self-isolation, and patient care [18]. The information at hand points to the impossibility of fully controlling COVID-19 infection in the nearish term. This fact suggests that creating guidelines to stop the spread of COVID-19 infection is essential for handling this public health issue. Therefore, the creation and observance of a set of rules and protocols will be crucial to the success of limiting this virus [19].

Given that the virus is usually transmitted through droplets and aerosols, healthcare workers, particularly dentists, are more likely to contract the illness. The symptoms of the illness and sensationalist articles in print, online, and other media may increase dentists’ anxiety about contracting the infection. Fear of the illness affects clinical judgement, rational discrimination, and psychological well-being [20]. Furthermore, dental professionals have a tough time safeguarding themselves from the virus, because infection spreads through asymptomatic people [21]. Under these situations, the specialists are highly concerned about their health and safety as well as those of their family, colleagues, and other patients. On the above background, the present study investigated the fear aspect in dentists in the initiation of clinical practice in times of COVID-19.

## 2. Materials and Methods

### 2.1. Study Design

This cross-sectional study was conducted based on a questionnaire circulated over April 2021 until July 2021 after being approved by the Committee for Institutional Ethics. The reporting of this study is based on the Strengthening the Reporting of Observational Studies in Epidemiology (STROBE) Statement. The questionnaire was validated and distributed online through emails among dental practitioners due to the lockdown being enforced in India by random sampling using the data pool acquired from Indian Dental Association (IDA) database (registered Indian dentists). 

The inclusion criteria followed was that the participants had to be dental practitioners, regardless of their specialization in dentistry, living in India and have an active email account. Other healthcare professionals, the general public, undergraduate and postgraduate students, candidates who gave incomplete data, and those not consenting, were excluded from the study. The study intent was communicated to all qualifying participants and the informed consent was labeled obligatory in the questionnaire. Sample size was estimated using the formula n = [DEFF*Np(1 − p)]/[(d2/Z21 − α/2 ∗ (N − 1) + p ∗ (1 − p)] and at 95% Confidence Level, with a minimum of 172 participants required to respond, attaining 80% power of the study.

### 2.2. Questionnaire 

The questionnaire was formulated in English and consisted of a systematic multiple-choice question, which was subsequently formatted into Google forms, a software used to collect information from people via personalized quizzes or surveys. The link was generated and circulated among the participants. The name and e-mail ID were created as required data to avoid discrimination more than once for an individual filling out the form. Two-step process of validation was conducted using two groups of experts (content validity index: 0.8) [22]. To assess the reliability, a pilot study on 30 dentists was conducted before the final study, which analyzed Cronbach’s alpha and split-half reliability values. The modifications were conducted based on results, and the final questionnaire had values of Cronbach’s alpha and split-half reliability of 0.80 and 0.81, respectively. The questionnaire had two sections. The first section was comprised of participants’ demographic characteristics such as place and country, the specialty of dentistry, gender, qualification, age, marital status, participants living arrangement, experience, type of institution they serve, and outlook regarding practice during COVID-19. The second section comprised 20 questions about the fear of practitioners relating to their professional and personal difficulties faced during COVID-19.

### 2.3. Data Analysis 

The collected information was transferred into an Excel spreadsheet and then imported into Statistical Package for Social Sciences (SPSS) version 25. Output measures were portrayed as simple frequency (n) and percentage (%), and the level of outcome measures were expressed as a mean and standard deviation (SD) for both demographic and subject–specific questions. To detect the significant difference between the different mean levels of impact among academic institutions and freelancing, the Chi-square test was used. A *p*-value of 0.05 or less was considered statistically significant.

## 3. Results

Description of Results

The data obtained, which was acceptable and complete according to the above-mentioned selection criteria, were 271 of 1000 email invitations sent that exhibited a response rate = 27.1%, deeming it more accurate [23]. Participants involved were 40.6% (110) from the city. Most of the participants specialized in the field of conservative dentistry and endodontics (25.50%). A total of 216 of the participants reported being a higher degree—Ph.D./Fellowship (79.7%). Data demonstrated that there was almost equal participation of male (49.1%) and female (50.9%) dental practitioners, of which many lived with spouses and family (50.2%). In the study, 142 of the participants were aged between 31–45 years (52.4%); 91 were 25–30 years (33.60%). When the number of years in the profession were taken into account, 86 were 11–20 years (31.70%); 73 were 6–10 years (26.90%); 71 were <5 years (26.20%) and 41 were >20 years (15.10%) from both academic (57.9%) and private practice (42.1%). Additionally, 218 (80.4%) of the individuals began practicing medicine after the first wave of COVID-19 ended, while 19.60% did not (Table 1).

When percent reduction practice was calculated (compared to before the COVID-19 outbreak in India in March 2020), 40 participants (14.8%) had no decrease in patient population, as they were in clinical practice due to COVID-19. A total of 57 (21%) experienced less than 10% decrease in practice whereas 95 (35.1%) had 21–40% reduction and 56 (20.70%) had 40–60%reduction. The salary was reduced for 43.5% and continued similarly for 56.5%. The death rate in their family or friends circle occurred for 37.10% of participants. It is interesting to note that 90.40% of practitioners were vaccinated with both doses and 9.60% with one dose with none without vaccination. When the participants were asked to rate the second wave compared to the first wave, 181 (66.8%) participants rated it extremely bad and 25.80% rated it very bad (Table 1).

When enquired about the stress faced by the practitioners, during the treatment of an individual in emergencies (such as trauma involving the face and oral region) compared to before the corona pandemic (before 2019), 167 (61.6%) participants agreed that it was stressful. Similarly, when asked about the stress of balancing the care for patients against concerns of contracting COVID-19 and spreading the virus to my family, colleagues, and other patients, 141 (52%) agreed and 115 (42.4%) strongly agreed about their stress on the spread as they might be carriers to their surroundings. Moreover, as a professional when they were asked about their fear of getting sued for the delay in initiating treatment during the pandemic,109 (40.2%) agreed with it and 128 (47.2%) disagreed with it because most practitioners know that in the pandemic very few people come to dentists, also most are in an academic set up where the institution will safeguard them. They are also apprehensive abour whether wearing an N95 respirator will completely protect them from inhaling droplet infection generated during procedures, as 162 (59.8%) agreed. The doubt of whether hand washes and fumigation will eliminate the virus comes to mind, which was felt by 150 (55.4%).

It was extremely stressful for professionals who have concerns about keeping their families safe, while still being able to provide quality care to their patients as agreed by 182 (67.2%) participants, as the risk of their families being exposed was likely. Moreover, when they were questioned about the stress professionals undergo to decide on whether to take a biopsy of abnormal tissue in the oral cavity immediately or delay it for some time, 164 (60.5%) of the participants agreed that it was stressful, as the fear of becoming exposed comes into play again. As a professional dilemma, 52.4% of participants are troubled by thoughts on what to do if they develop an upper respiratory tract infection, whether to tell family and colleagues who may get unduly worried and self-quarantine until the COVID-19 test report arrives. As a professional to decide on whether to start an emergency procedure (such as tracheotomy, RCT, or foreign body retrieval) in the time of the pandemic is faced by 58.7% of the practitioners, as these emergency procedures involve aerosol production, which is a likely mode of transmission for virus particles (Table 2).

The professionals are also worried about whether they should screen all patients entering the clinic/hospital and whether they should insist on a COVID-10 test report before treating them during the pandemic period for their safety, as it would further create an impact on their families. Similarly, it is a matter of concern to think about the adequate quantity and quality of personal protective equipment (PPE) at work; which is the concern for about 163 (60.1%) participants. On an interesting note, most people (52.4%) agreed that their belief in God and his protection has strengthened. Apprehensive as to vaccination will prevent from getting infected and developing severe issues was there for most of the participants, at 151 (55.7%), which is due to the reports being published recently (Table 2).

When questioned about whether the recent critical situations due to COVID-19 keep coming to mind and affecting their work, most people 137 (51.3%) answered positively that it affects them only sometimes. On the other hand, 50% of the participants had sometimes felt that they have completely lost their personal and social aspects that made them happy or gave them peace of mind. Moreover, sometimes they feel like keeping a distance from people, and avoiding the interaction makes them irritated (47.3%). The fear of making future decisions (47%), neglecting patients who need help (51.5%), anticipation of earning for the next few months (46.2%), and finally, the worry of becoming sick and putting their family at risk (38.6%) comes into their thoughts only sometimes (Table 2).

When the Chi-square test was used to compare the demographic data as well as subject-specific questions such as participants working in an academic institution and private practice, there was no significant difference in both except in a few categories such as the specialty of dentistry, where most of the freelancers were Pedodontists (*p*-value = 0.005); most of the freelancers was married (*p*-value = 0.065); clinical practice after the first wave mostly was by freelancers (*p*-value = 0.01); 19.1% attached to institutes did not have changes in earnings; 28.1% and 39.5% of private practitioners had a maximum of 40% reduction. Almost a quarter of the institution staff had 40 to 60% reduction (*p*-value = 0.002); 28.1% freelancers had less than 10% cut and 33% of institution staff had 40 to 60% reduction in salary (*p*-value = 0.006); 57% private practitioners have a fear of being sued for the delay in treatment (*p*-value = 0.036); and 62% private practitioners have apprehensions about the vaccine being preventive (*p*-value = 0.026). In the study, 33% working in institutions fear getting sick and putting their family at risk (Table 2).

## 4. Discussion

The current health crisis brought on by the coronavirus has produced major worries, stress, and anxiety among several populations [24]. A pandemic that spreads over the world can cause a lot of anxiety. The infodemic is an additional component of COVID-19 issues. The infodemic is an overabundance of information, both genuine and deceptive, that overstimulates the subject by impairing its capacity for adequate information processing [25]. Concerns about a person’s wellbeing, such as the excessive use of social media and severe psychological distress, are linked to the COVID-19 fear paired with the infodemic [26]. Fear of dental professionals becoming infected has grown as a result of SARS-high CoV2’s infectious potential and their exposure to the virus due to their frequent interaction with bodily fluids, which is the main method of transmission [20,27]. Thus, the organism spreading from the air and being inhaled is a big apprehension in such cases. Many organizations, including the Centers for Disease Control and Prevention (CDC), American Dental Association (ADA), British Dental Association, and National Health Service, have created response groups and guidelines for dental settings, regardless of how the infection spreads, in response to the current pandemic.

Dental professionals could very well get the infection from patients and spread it to their surroundings because the virus can spread from asymptomatic carriers. Similar to this study, a study that evaluated dentists in almost 30 countries found that 90% of professionals worry about contracting an infection from clients or coworkers [28]. The majority of the professionals treating these patients are mostly working in academic institutions and 31.7% have 11–20 years of experience in the profession. Although the dental team is adequately protected by personal protective equipment (PPE), N95 masks, and face shields, the majority of the study populations still worry about contracting an infection while wearing PPE.

Numerous medico-legal difficulties regarding patient care provisions for Indian dentists were brought up by the COVID-19 pandemic. As a result, the Indian government and the Dental Council of India (DCI) published guidelines on acceptable patient treatment procedures during a pandemic, where 75.5% of dentists expressed concern about being sued as a result of the rules’ resemblance to the research [29]. Due to recent studies that have been published, the majority of participants are concerned about immunizations preventing the spread of the condition [30]. Although the proportion of the populace that must receive vaccinations in order to develop herd immunity against COVID-19 is still unknown, in general, 50–90% of the population must be protected, either naturally or through vaccination, in order to develop herd immunity [30]. On the other hand, following COVID-19 vaccination, the likelihood of hospitalization is 0.06% [31]. Polls conducted by Dental Post/RDH magazine in 2019 and 2020 demonstrate that COVID-19 did not cause any difficulties for dental employment. As the majority of practitioners did not see a decrease in their practice, it exposed them more and put pressure on them, leaving gaps that need to be filled up. The individuals’ intention to become immunized was significantly negatively impacted by a number of distinctive characteristics. Investigations carried out in France [32], China [33], and Europe [34], all revealed decreased vaccination willingness in female participants. Recognizing these factors could aid in creating population-focused awareness efforts that would enhance immunization rates. There was a tendency among Jordanians to believe in the COVID-19 conspiracy ideas [30]. The careful sharing of medical information, social media campaigns, and creating a culture of fact-checking are a few tactics that have been proposed to counteract conspiracy theories.

Additional key findings are that many dental practitioners have been mentally affected because of the recent critical crisis, as we are in the high-risk category. Dentists have an ethical obligation to give emergency dental care to patients who require immediate action, regardless of their health. Patient care should not be overlooked, according to 35.5% of dentists, because it is the dentist’s moral and ethical obligation to balance patients’ demands with public health issues. Moreover, the mindset and willingness to interact with people have come down, as they have completely lost their social aspects of life, according to 50% of dentists.

As COVID-19 cases again surge across India, dentists were apprehensive about the restrictions they experienced during the previous wave in 2020, which resulted in fear about their future and financial issues. However, as of now, no states were asking dental offices to limit essential services, and dentists are optimistic about being able to continue dental care amidst the pandemic. All these stressors bog down the dentists only sometimes, suggesting that the study population is quite positive and is not bogged down very much. As dentists continue to work their way through this pandemic, determining how to provide necessary dental care within the context of patients’ needs, area-specific COVID-19 disease burden, PPE supply, and the health care capabilities and capacity is necessary to overcome all these fears. In many countries around the world, vaccine reluctance still exists [35,36,37,38,39,40]. The World Health Organization acknowledges that one of the greatest health dangers today is vaccination reluctance. The COVID-19 vaccinations may offer a defense against the Alpha, Gamma, Beta, and Delta versions, according to the evidence. Fear of the COVID-19 vaccine is currently an important problem. Additionally, social bonding activities can encourage information sharing about the alleged dangers of the COVID-19 vaccines, which might increase vaccine reluctance. Consequently, persons who are worried about the vaccination’s negative effects may not accept the COVID-19 vaccine as well. To convince individuals to seek vaccines, however, is a barrier that policymakers, healthcare officials, and experts must overcome [41]. Personal face-to-face interactions have been linked to a variety of positive effects on health. Social involvement and bonding can help people cope with the pandemic by fostering trust in the COVID-19 vaccinations and so promote vaccine uptake [41]. The vaccines developed against COVID-19 are reported to be decreasing the spread and severity of SARS-CoV-2 viral infections [33,42]. In the current study, most of the respondents were vaccinated with two doses of COVID-19. It will take deliberate effort to clarify the risks and benefits of various vaccinations so that the population may comprehend the reasoning behind vaccine uptake recommendations in the direction to address the lack of public confidence in vaccines. Being beneficial to increase public trust in the COVID-19 immunization program, greater efforts must be made to communicate the advantages and hazards using evidence-based data. Furthermore, social bonding therapies will be essential to somehow become ready for a post-pandemic environment, where strengthening existing relationships or creating new ones would likely be crucial for helping people cope with challenges that they endured during COVID-19 [41].

A Recent national survey found that it was observed that dentists increased their propensity for using telehealth. The dentists believed that online education had become more popular. Moreover, during the pandemic, some dentists considered leaving the field [21]. In a study of Australian residents, Armfield et al. [43] found that those with high fear were more likely to have a longer time since their last dental appointment. Pohjola et al. [44] found that among Finnish individuals, frequent dental visits were linked to high levels of dread. Similar findings were made in Iran by Saatchi et al. [45], who found that irregular attendees were more fearful than regular ones. This avoidance of dental care due to anxiety may aggravate pre-existing oral diseases, start a vicious cycle of fear, increase treatment requirements, and worsen oral health status [46].

A cross-sectional questionnaire study is accompanied with numerous biases. The study relied on self-reporting and may reflect under- or over-reporting. Additionally, because the study was conducted online, it was unable to specifically target clinicians who were uncomfortable with the online format or had almost no online presence. The close ended nature of the study’s questionnaire might have missed significant unavailable responses or failed to accommodate for the wide ranges of response. Moreover, a clinical validating scale and multiple comparison could have been responsible for more inferring results. Furthermore, confirmation, social desirability, and central tendency biases cannot be taken into account in such an online survey.

## 5. Conclusions

New cases of the COVID-19 pandemic are still being reported, and because there is a risk of infection spreading from patients to dental care workers, further safety precautions are needed to prevent the spread of the COVID-19 pandemic. The current survey concluded that COVID-19 pandemic’s impact on humanity makes dentists nervous and afraid, despite their high level of knowledge and practice in their diverse specialties. It also affected dentists’ practices, as per the statistics presented. Dentists felt a great deal of fear during the COVID-19 pandemic, with a sizable portion displaying some anxiety towards spreading the infection to their family. Dentists’ financial situation was impacted as well, and it could be worthwhile to look into its timeframe, incentive plans, and potential future infrastructure needs.

## Figures and Tables

**Table 1 vaccines-10-02081-t001:** Demographic, family, and professional details of the dental professionals who volunteered in the study.

Questions	Choices	Count	Percent
Place	Town	63	23.20
	City	110	40.60
	Metro	98	36.20
Specialty of Dentistry	General Dentist	56	20.70
	Conservative Dentist	69	25.50
	Oral Medicine and Radiology	39	14.40
	Prosthodontics	22	8.10
	Oral Pathologist	6	2.20
	Orthodontics	15	5.50
	Pedodontist	31	11.40
	Oral and Maxillofacial Surgeon	23	8.50
	Community dentist	10	3.70
Educational qualification	Undergraduate	43	15.90
	Postgraduate	216	79.70
	Higher Degree (PhD/Fellowship)	12	4.40
Gender	Female	138	50.90
	Male	133	49.10
Marital Status, 3 choices	Single	88	32.50
	Married	181	66.80
	Widow/Widower	2	0.70
Marital Status, 2 choices (divorce merged)	Single	88	32.50
	Married/Widow/Widower	183	67.50
Age	25–30 years	91	33.60
	31–45 years	142	52.40
	>45 years	38	14.00
Your living arrangement	Living in Hostel or PG accommodation or Hotel	54	19.90
	Living with family (spouse and kids)	136	50.20
	Living with extended family (Parents/in laws, spouse and children)	81	29.90
Number of years in the profession:	<5 years	71	26.20
	6–10 years	73	26.90
	11–20 years	86	31.70
	>20 years	41	15.10
Type of institution you are serving:	Dental College/Medical College/Academic institution	157	57.90
	Freelancing/visiting on call/Private practice in clinic	114	42.10
Did you start your clinical practice after the end of first wave of COVID-19?	Yes	218	80.40
	No	53	19.60
If you did start clinical practice after the end of first wave of COVID-19, what was the percent reduction in your practice when compared to before COVID-19 outbreak in India in March 2020: 5 choices	No decrease (as have not been in to clinical practice)	40	14.80
	Less than 10 decrease in practice when compared to before March 2020	57	21.00
	21 to 40 decrease in practice when compared to before March 2020	95	35.10
	40 to 60 decrease in practice when compared to before March 2020	56	20.70
	More than 60 decrease in practice when compared to before March 2020	23	8.50
If you did start clinical practice after the end of first wave of COVID-19, what was the percent reduction in your practice when compared to before COVID-19 outbreak in India in March 2020: 4 choices	No decrease (as have not been in to clinical practice)	40	14.80
	Less than 10 decrease in practice when compared to before March 2020	57	21.00
	21 to 40 decrease in practice when compared to before March 2020	95	35.10
	40 to 60 decrease in practice when compared to before March 2020	79	29.20
	More than 60	0	0.00
Did you have a salary cut last year due to COVID-19 pandemic	Yes	103	43.50
	No	134	56.50
Did anyone in your family or friend circle close to you died due to COVID-19	Yes	96	37.10
	No	163	62.90
Have you taken a vaccine for COVID-19	No	0	0.00
	Only one dose	26	9.60
	Yes both doses	245	90.40
When compared to first wave of COVID-19 last year how will you rate the COVID-19 this year	Same as last year	4	1.50
	Moderately bad	16	5.90
	Very bad	70	25.80
	Extremely bad	181	66.80

**Table 2 vaccines-10-02081-t002:** Chi square test to compare the question 10 with all the other questions.

	Categories	N	Type of Institution You Are Serving:		Chi Square *p* Value
			Academic Institution N (%)	Freelancing//Private Practice N (%)	
1. When compared to before the Corona pandemic (before 2019), is treating an individual in emergency situations (such as trauma involving face and oral region) highly stressful?	Strongly Disagree	9	5 (3.2)	4 (3.5)	0.462 0.927
	Disagree	30	17 (10.8)	13 (11.4)	
	Agree	167	95 (60.5)	72 (63.2)	
	Strongly Agree	65	40 (25.5)	25 (21.9)	
2. Is it very stressful to balance caring for patients against concerns of contracting COVID-19 and spreading the virus to family, colleagues and other patients?	Strongly Disagree	6	5 (3.2)	1 (0.9)	2.085 0.555
	Disagree	9	6 (3.8)	3 (2.6)	
	Agree	141	79 (50.3)	62 (54.4)	
	Strongly Agree	115	67 (42.7)	48 (42.1)	
3. As a professional, do I fear that I might get sued for any delay in initiating treatment due to the COVID-19 pandemic situation?	Strongly Disagree	18	11 (7)	7 (6.1)	8.516 **0.036 ***
	Disagree	128	63 (40.1)	65 (57)	
	Agree	109	74 (47.1)	35 (30.7)	
	Strongly Agree	16	9 (5.7)	7 (6.1)	
4. Am I apprehensive as to whether wearing an N95 respirator will completely protect me from inhaling droplet infection generated during procedures?	Strongly Disagree	15	11 (7)	4 (3.5)	3.679 0.298
	Disagree	53	26 (16.6)	27 (23.7)	
	Agree	162	94 (59.9)	68 (59.6)	
	Strongly Agree	41	26 (16.6)	15 (13.2)	
5. At times, do I have doubts as to whether hand wash and fumigation will really eliminate the virus?	Strongly Disagree	22	15 (9.6)	7 (6.1)	3.526 0.317
	Disagree	57	29 (18.5)	28 (24.6)	
	Agree	150	85 (54.1)	65 (57)	
	Strongly Agree	42	28 (17.8)	14 (12.3)	
6. Is it extremely stressful for professionals who have concerns about keeping their families safe, while still being able to provide quality care to their patients?	Strongly Disagree	5	4 (2.5)	1 (0.9)	2.271 0.518
	Disagree	7	5 (3.2)	2 (1.8)	
	Agree	182	101 (64.3)	81 (71.1)	
	Strongly Agree	77	47 (29.9)	30 (26.3)	
7. Is it very stressful for professionals to make a decision on whether to take a biopsy of abnormal tissue in the oral cavity immediately or delay it for some time?	Strongly Disagree	9	4 (2.5)	5 (4.4)	2.59 0.459
	Disagree	67	41 (26.1)	26 (22.8)	
	Agree	164	91 (58)	73 (64)	
	Strongly Agree	31	21 (13.4)	10 (8.8)	
8. As a professional, I am troubled by thoughts on what to do if I develop an upper respiratory tract infection. Should I tell family and colleagues (who may get unduly worried) and self-quarantine until the COVID-19 test report arrives?	Strongly Disagree	15	10 (6.4)	5 (4.4)	1.82 0.611
	Disagree	44	24 (15.3)	20 (17.5)	
	Agree	142	86 (54.8)	56 (49.1)	
	Strongly Agree	70	37 (23.6)	33 (28.9)	
9. Is it stressful for me as a professional to take a decision on whether to start an emergency procedure (such as tracheotomy, RCT, foreign body retrieval) in the time of the pandemic?	Strongly Disagree	14	8 (5.1)	6 (5.3)	1.92 0.589
	Disagree	59	30 (19.1)	29 (25.4)	
	Agree	159	94 (59.9)	65 (57)	
	Strongly Agree	39	25 (15.9)	14 (12.3)	
10. Am I worried whether we should screen all patients entering the clinic/hospital and whether we should insist on a COVID-19 test report before treating them in the pandemic period?	Strongly Disagree	6	3 (1.9)	3 (2.6)	2.693 0.441
	Disagree	45	22 (14)	23 (20.2)	
	Agree	161	94 (59.9)	67 (58.8)	
	Strongly Agree	59	38 (24.2)	21 (18.4)	
11. It is a matter of concern for me to think whether we have adequate quantity and quality of personal protective equipment (PPE) at work.	Strongly Disagree	7	3 (1.9)	4 (3.5)	5.985 0.112
	Disagree	39	21 (13.4)	18 (15.8)	
	Agree	163	89 (56.7)	74 (64.9)	
	Strongly Agree	62	44 (28)	18 (15.8)	
12. Compared to before these experiences, my spiritual faith has strengthened.	Strongly Disagree	12	6 (3.8)	6 (5.3)	0.505 0.918
	Disagree	59	33 (21)	26 (22.8)	
	Agree	142	84 (53.5)	58 (50.9)	
	Strongly Agree	58	34 (21.7)	24 (21.1)	
13. I am apprehensive whether vaccination will prevent me from getting infected and developing severe issues.	Strongly Disagree	20	10 (6.4)	10 (8.8)	9.23 0.026 *
	Disagree	79	49 (31.2)	30 (26.3)	
	Agree	151	80 (51)	71 (62.3)	
	Strongly Agree	21	18 (11.5)	3 (2.6)	
14. The recent critical situations due to COVID-19 keeps coming in mind and is affecting my work.	Not at all	19	9 (5.9)	10 (8.8)	4.911 0.178
	Some times	137	87 (56.9)	50 (43.9)	
	Most times	69	37 (24.2)	32 (28.1)	
	Always	42	20 (13.1)	22 (19.3)	
15. I have completely lost personal and social aspects that made me happy or gave me peace of mind.	Not at all	48	25 (16.4)	23 (20.2)	2.335 0.506
	Some times	133	74 (48.7)	59 (51.8)	
	Most times	68	44 (28.9)	24 (21.1)	
	Always	17	9 (5.9)	8 (7)	
16. I feel I am keeping distance from people, avoid interacting with people and am irritated.	Not at all	60	32 (21.3)	28 (24.6)	1.763 0.623
	Some times	125	70 (46.7)	55 (48.2)	
	Most times	59	34 (22.7)	25 (21.9)	
	Always	20	14 (9.3)	6 (5.3)	
17. I feel I am neglecting many patients who need my help.	Not at all	93	48 (32.2)	45 (39.8)	1.871 0.6
	Some times	135	81 (54.4)	54 (47.8)	
	Most times	28	17 (11.4)	11 (9.7)	
	Always	6	3 (2)	3 (2.7)	
18. I have difficulty thinking and making decisions about future.	Not at all	56	29 (19.2)	27 (23.9)	3.064 0.382
	Some times	124	71 (47)	53 (46.9)	
	Most times	56	31 (20.5)	25 (22.1)	
	Always	28	20 (13.2)	8 (7.1)	
19. I am anticipating difficulty in earning for the next few months.	Not at all	25	14 (9.4)	11 (9.7)	0.71 0.871
	Some times	121	72 (48.3)	49 (43.4)	
	Most times	80	44 (29.5)	36 (31.9)	
	Always	36	19 (12.8)	17 (15)	
20. Worrying about getting sick and putting family to risk puts a lot of strain on me.	Not at all	11	2 (1.3)	9 (8)	9.039 **0.029 ***
	Some times	102	59 (39.1)	43 (38.1)	
	Most times	74	40 (26.5)	34 (30.1)	
	Always	77	50 (33.1)	27 (23.9)	

The values marked with (*) are statistically significant.

## Data Availability

The data that support the findings of this study are available from the corresponding author, upon reasonable request.

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
