# Peer review of "Apprehensions and Optimism among Dental Professionals during the COVID-19 Pandemic—A Nationwide Cross-Sectional Evaluation"

_vaccines, 2022, doi:10.3390/vaccines10122081_

Round 1
Reviewer 1 Report
This paper describes a fear of COVID-19 among dental practitioners. The questions were asked in questionnaire format via Google form. There were 271 participants who participated in this study. The authors used a basic Chi-square test to test the hypothesis regards to their research question. The results show that the dentists reported that they had stressed over getting exposed to the virus as well as spreading the virus to their families.
This paper relatively had very limited inferential power from many perspectives: sample size, question design, and suitability of statistic tools. My concerns are listed in the following
Major concern:
1. Sample size: The sample is relatively small. There are only 271 participants. To mitigate this issue, please add more detail on how well these participants represent your population group.
2. Question design: in “Questionnaire” (the line between 108 to 118). The authors failed to report on the question validation. The authors need to show the validation procedure or validation result or even have some citation of the source of these questions.
3. Suitability of statistic tools:
a. Regardless of sample size, the authors should have done better than this. The questionnaire they used could have been converted to an index or score. Later, those can be used as a continuous variable to gain more implications. Please check an example, the trust indexes were calculated from the Liker scale (Juarez et al.,2022). It is very similar to this case. if the authors successfully build the index, they can even develop more impactful research questions.
b. In the conclusion section, the authors need to describe more about how the statistic analysis leads to the conclusion that “Since the study population here is quite optimistic, the assurance of overcoming this fear can be brought about in a short span.” (line 270 to 271)
Minor concern:
1. Marking statistically significant should be used asterisk like *** represents P<0.01, ** represents P<0.05.etc. Please do not use just the word “Significant”. I don’t think it is the correct way
2. Table A1, the authors can add the p-value for the Chi-square test as well.
Juarez, R., Phankitnirundorn, K., Okihiro, M., & Maunakea, A. K. (2022). Opposing Role of Trust as a Modifier of COVID-19 Vaccine Uptake in an Indigenous Population. Vaccines, 10(6), Article 6. https://doi.org/10.3390/vaccines10060968
Author Response
We would like to thank the academic editor and the reviewers for taking out their precious time to review this manuscript and give us their comments. We would like to explicitly state that we agree with all the comments as these helped us improve the quality of our paper. We have made a conscious effort to answer all the remarks in the paper as advised by the reviewers and highlighted changes made in red for their convenience. Kindly consider these and excuse us for any lapse on our part.

Reviewer 2 Report
Although the paper appears fine, I suppose that the sample size is still low.
On the one hand, dentists who did not respond to the survey could be involved. On the other hand, you could also try to call in dentists from other countries, in order to increase the sample size.
It would also be appropriate to include, in the paper, more evidence of COVID-19 vaccine effectiveness also against symptomatic disease, in order to reassure those dentists who experienced uneasiness and concerns from the pandemic
Author Response

(The authors gave the same response as above.)

Reviewer 3 Report
Summative comments - I have read the manuscript with interest. The topic is of interest. However, I have several concerns about the manuscript, in its current state, and therefore cannot recommend publication.
1. The introduction does not provide an adequate rationale for the study question. Further details would have been desirable regarding the effect of the pandemic on stress as has been studied in other populations. Also essential would be a review of the pandemic in India, the timeline of case trends, and national/professional society guidance.
2. The methods section requires much improvement. Kindly refer to STROBE reporting guidelines for observational studies.
a. How was the study population identified?
b. Describe the sampling method in detail.
c. How was sample size calculated?
d. How was the questionnaire created? How was it validated?
e. Was the questionnaire piloted ? How were modifications made, if any?
f. What steps were taken to mitigate bias?
g. The survey instrument needs to be included in submissions.
h. Several statements in the Likert scale questionnaire are double-barreled which affects the reliability of the results.
i. Why was a clinically validated scale to measure stress not utilized when there are several such scales available?
j. Addition of multivariate analysis would be desirable.
k. Was a correction for multiple comparison problem considered?
3. Results - The response rate needs to be included in reporting of surveys. The demographics of the respondents should be compared to known demographics of the population of interest, i.e. dentists in India, to assess if study population is comparable, especially if nonrandom sampling is utilized.
Author Response

(The authors gave the same response as above.)

Reviewer 4 Report
This is a highly interesting article and has much relevance and potential to attract good readership. The authors need to further review relevant COVID-19 literature addressing some of the points they discuss e.g. Ramkissoon, H. (2021). Social bonding and public trust/distrust in COVID-19 vaccines. Sustainability, 13(18), 10248.
Thank you.
Author Response
Response to Reviewer 4 Comments
The changes can be found in the revised version in RED.
Point 1: Comments and Suggestions for Authors
This is a highly interesting article and has much relevance and potential to attract good readership. The authors need to further review relevant COVID-19 literature addressing some of the points they discuss e.g. Ramkissoon, H. (2021). Social bonding and public trust/distrust in COVID-19 vaccines. Sustainability, 13(18), 10248.
Response 1: Thank you so much for the positive comment. We have added relevant content in the introduction as well as the discussion part. Hopefully now acceptable.

Round 2
Reviewer 1 Report
I appreciate the effort that you put into making the revision as I suggested.
The quality of the paper has significantly improved. I think the paper has met the publication requirement of the journal.
Best regards
Author Response
Response to Reviewer 1 Comments
Point 1: I appreciate the effort that you put into making the revision as I suggested.
Response 1: Thank you so much for the positive comment.
Point 2: The quality of the paper has significantly improved. I think the paper has met the publication requirement of the journal.
Response 2: Thank you for the encouraging comment towards our paper. We highly appreciate it.

Reviewer 3 Report
I thank the authors for their careful consideration of suggestions and comments. Some notable improvements have been made in the description of the study methodology. However, several concerns remain. I have previously stated my concerns regarding the appropriateness of chosen study methodology to the study question so will not restate them. I appreciate the authors' acknowledgment of the same in the limitations of the study. However, that still limits the validity of the findings and therefore the utility of the study itself.
1. Introduction - While the authors acknowledge the comment from the previous review in their response, no changes have been made to the introduction. The introduction, as it stands, does not provide an adequate rationale for the current study.
In addition, the introduction contains several statements which have not been supported with proper citations. Some examples include - Line 57-59; 63-64; 82-84;87-89. Please ensure that when a statement is made stating a fact, it is appropriately supported.
2. Methods - Methods section is much improved. Please specify in the manuscript that random sampling was done using the data pool acquired from IDA database.
3. Discussion - The discussion section needs extensive editing of language to clarify the message. For example, line 259-261, " Given that personal ...... provide adequate protection.... most of the study population have fear of getting infected even after wearing personal protective equipment" The sentence has an inherent contradiction.
The discussion section also needs a much more robust discussion of existing literature, including data on pandemic stress in India in the general population, other healthcare professionals, and pandemic stress in dentists in other nations - with an examination of factors contributing and how it relates to the results of the current study.
Limitations section, while somewhat improved, still does not address all the potential biases in this study. To name a few, sampling bias - questionnaire in online format excludes those without web presence/ limited comfort with online format, same for English language proficiency; Confirmation bias - given the nature of questions asked; Social desirability bias; Central tendency bias; etc.
4. Conclusion - The authors' conclusion are not supported by results. Please revise this. For example, authors make a conclusion about "dentists across the world..." while the sample is a very small sample of dentists in India.
Author Response
Response to Reviewer 2 Comments
Point 1: I thank the authors for their careful consideration of suggestions and comments. Some notable improvements have been made in the description of the study methodology. However, several concerns remain. I have previously stated my concerns regarding the appropriateness of chosen study methodology to the study question so will not restate them. I appreciate the authors' acknowledgment of the same in the limitations of the study. However, that still limits the validity of the findings and therefore the utility of the study itself.
Response 1: Thank you so much for the positive comment and appreciating our revised manuscript. We have tried and addressed to all the constructive comments raised.
Point 2: Introduction - While the authors acknowledge the comment from the previous review in their response, no changes have been made to the introduction. The introduction, as it stands, does not provide an adequate rationale for the current study.
Response 2: Thank you for raising this concern. In the revised manuscript the introduction section is now added for the background for conducting the current study with addition of 3 references. Hopefully now acceptable.
Point 3: Line 302 Table 3 – It seems only the result of ANOVA. Please describe Post hoc Tukey’s test.
Response 3: Thank you for raising this concern. We have now made a new table with all the required data. Hopefully now acceptable.
Point 4: In addition, the introduction contains several statements which have not been supported with proper citations. Some examples include - Line 57-59; 63-64; 82-84;87-89. Please ensure that when a statement is made stating a fact, it is appropriately supported.
Response 4: We sincerely apologise for this gross error. We have now revised the same and added references where suggested and required. Hopefully now acceptable.
Point 5: Methods - Methods section is much improved. Please specify in the manuscript that random sampling was done using the data pool acquired from IDA database.
Response 5: Thank you respected reviewer for this constructive comment. We have added same in the revised manuscript. Hopefully now acceptable.
Point 6: Discussion - The discussion section needs extensive editing of language to clarify the message. For example, line 259-261, " Given that personal ...... provide adequate protection.... most of the study population have fear of getting infected even after wearing personal protective equipment" The sentence has an inherent contradiction.
Response 6: Thank you for the valuable input. We have now rephrased the sentences where required in the discussion section. Hopefully now acceptable.
Point 7: The discussion section also needs a much more robust discussion of existing literature, including data on pandemic stress in India in the general population, other healthcare professionals, and pandemic stress in dentists in other nations - with an examination of factors contributing and how it relates to the results of the current study.
Response 7: We have now addressed the suggested comment in the discussion section with additional references.
Point 8: Limitations section, while somewhat improved, still does not address all the potential biases in this study. To name a few, sampling bias - questionnaire in online format excludes those without web presence/ limited comfort with online format, same for English language proficiency; Confirmation bias - given the nature of questions asked; Social desirability bias; Central tendency bias; etc.
Response 8: Thank you for the valuable input. We have now addressed the suggested comment in the limitations.
Point 9: Conclusion - The authors' conclusion are not supported by results. Please revise this. For example, authors make a conclusion about "dentists across the world..." while the sample is a very small sample of dentists in India.
Response 9: We have now addressed the suggested comment revised the conclusion. Hopefully now acceptable.
